# Chronic Low Dose Neutron Exposure Results in Altered Neurotransmission Properties of the Hippocampus-Prefrontal Cortex Axis in Both Mice and Rats

**DOI:** 10.3390/ijms22073668

**Published:** 2021-04-01

**Authors:** Balaji Krishnan, Chandramouli Natarajan, Krystyn Z. Bourne, Leila Alikhani, Juan Wang, Allison Sowa, Katherine Groen, Bayley Perry, Dara L. Dickstein, Janet E. Baulch, Charles L. Limoli, Richard A. Britten

**Affiliations:** 1Mitchell Center for Neurodegenerative Diseases, Department of Neurology, The University of Texas Medical Branch, Galveston, TX 77555, USA; chnatara@utmb.edu (C.N.); kzbinoh@hotmail.com (K.Z.B.); 2Department of Radiation Oncology, University of California, Irvine, CA 92697, USA; lalikhan@uci.edu (L.A.); jbaulch@hs.uci.edu (J.E.B.); climoli@uci.edu (C.L.L.); 3Department of Pathology, Uniformed Services University of Health Sciences, 4301 Jones Bridge Rd., Bethesda, MD 20814, USA; jun.wang.ctr@usuhs.edu (J.W.); katherine.groen.ctr@usuhs.edu (K.G.); Bailey.perry.ctr@usuhs.edu (B.P.); dara.dickstein.ctr@usuhs.edu (D.L.D.); 4The Henry M. Jackson Foundation for the Advancement of Military Medicine (HJF), 6720A Rockledge Drive, Bethesda, MD 20817, USA; 5Microscopy CoRE and Advances Bioimaging Center, Icahn School of Medicine at Mount Sinai, 1468 Madison Ave., New York, NY 10029, USA; Allison.sowa@mssm.edu; 6Department of Radiation Oncology, Eastern Virginia Medical School, Norfolk, VA 23507, USA; brittera@evms.edu

**Keywords:** space radiation, neutrons, charged particles, dendritic spines, myelin, synapses, long-term depression, C57Bl/6 mice, Wistar rats, FASS-LTP

## Abstract

The proposed deep space exploration to the moon and later to Mars will result in astronauts receiving significant chronic exposures to space radiation (SR). SR exposure results in multiple neurocognitive impairments. Recently, our cross-species (mouse/rat) studies reported impaired associative memory formation in both species following a chronic 6-month low dose exposure to a mixed field of neutrons (1 mGy/day for a total dose pf 18 cGy). In the present study, we report neutron exposure induced synaptic plasticity in the medial prefrontal cortex, accompanied by microglial activation and significant synaptic loss in the hippocampus. In a parallel study, neutron exposure was also found to alter fluorescence assisted single synaptosome LTP (FASS-LTP) in the hippocampus of rats, that may be related to a reduced ability to insert AMPAR into the post-synaptic membrane, which may arise from increased phosphorylation of the serine 845 residue of the GluA1 subunit. Thus, we demonstrate for the first time, that low dose chronic neutron irradiation impacts homeostatic synaptic plasticity in the hippocampal-cortical circuit in two rodent species, and that the ability to successfully encode associative recognition memory is a dynamic, multicircuit process, possibly involving compensatory changes in AMPAR density on the synaptic surface.

## 1. Introduction

NASA’s planned inter-planetary explorations has increased the need for information on the health risks associated with prolonged exposure to deep space flight stressors, including space radiation (SR) exposure. The deep space radiation spectrum of Galactic Cosmic Radiation (GCR) is composed of highly energetic, high mass (Z ≤ 26) charged particles. Current estimates suggest that astronauts will be exposed to ~13 cGy of SR during each year of the mission [1], the majority of which will be incurred while in transit to Mars. The structure of the spacecraft will offer a degree of shielding to the astronauts, reducing the SR dose and altering the GCR spectrum from that seen in free space. The “Local-Field” spectrum (i.e., the SR spectrum that the internal organs of astronauts will receive within the spacecraft) calculated using the current spacecraft design specifications predicts that the majority of the physical and equivalent SR dose will arise from Z < 15 particles [2,3].

There is an increasing body of evidence from ground-based rodent experiments that exposure to <25 cGy of several component SR ions (protons, ^4^He ^16^O, ^28^Si, ^48^Ti and ^56^Fe) impairs various aspects of executive function [4,5,6,7,8,9,10,11,12,13,14]. There was some concern that these studies using single bolus exposures delivered at a dose rates several orders of magnitude greater than that predicted for GCR exposures were potentially over-estimating the risk of SR-induced cognitive impairment. A unique study was thus undertaken at the Colorado State University ^252^Cf facility that provides a continuous source of neutrons at a dose rate of 1 mGy per day with photons contributing 20% of the total dose, and that serves as a proxy for some aspects of the high LET component of SR [15]. Cohorts of mice and rats were simultaneously exposed to 18 cGy neutrons over a 180-day period, then returned to the investigators’ institutes for assessments of behavioral performance using one of three cognitive flexibility tasks. For mice, the Object-In-Place (OIP) test was used [4], and for rats Attentional Set Shifting (ATSET) and the Unconstrained cognitive flexibility (UCFlex) tasks were used [16]. Despite the different species and different cognitive flexibility tasks, there was a high degree of similarity in the performance decrements seen in the neutron-exposed rodents. Using Kernel Density Estimation (KDE) of the probability density function of the cognitive performance metrics, the numbers of animals within sham and neutron-exposed cohorts that exhibited severe cognitive performance impairment were established. The level of performance that was considered to represent severe impairment was set at the 5th percentile of the sham cohort performance profile (conceptually analogous to a Z-score of 2). KDE analyses revealed that relative to the 5% impairment threshold in control rats, 41% of neutron-exposed rats had severely compromised performance in the simple discrimination (SD) stage of the ATSET test [16], while 35% of neutron-exposed mice had severely compromised performance in the OIP test [4]. These values were highly consistent with the ~44% of rats who had impaired ATSET performance after exposure to a bolus (high dose rate) of ^56^Fe particles [10]. These data collectively suggest that there may be a common mechanistic basis for the neutron-induced loss of cognitive performance in the OIP and the SD stage of the ATSET testing paradigms. The SD stage interrogates the rats’ decision-making abilities and requires the subject to make an association between a specific stimulus (cue) and the presence of a food reward, i.e., associative recognition memory. Performance in the OIP task also requires a high degree of associative memory formation, specifically where an object was located, that information then serves as a reference memory for later aspects of the task. Importantly, performance in both the SD stage of ATSET and OIP task would be severely compromised if there were defects in the encoding or retrieval of associative recognition memory.

OIP associative recognition memory depends on two hippocampal-cortical circuits, involving the hippocampus (HPC), perirhinal (PRH) and medial prefrontal (mPFC) cortices [17]. Encoding of OIP associative recognition memory is highly dependent upon N-methyl-D-aspartate receptors (NMDAR) within the hippocampus [17]. Further, it would seem likely that the encoding of the SD food cue required for associative recognition memory would be similarly regulated.

In the context of network level “learning” responses, both long-term potentiation (LTP) and long-term depression (LTD) work in unison to allow repeated “consolidation” of memory, by appropriately regulating synaptic strength so that reinforcement is possible [18]. This process is referred to as homeostatic synaptic plasticity (HSP), which is a negative feedback mechanism that neurons use to offset excessive excitation or inhibition by adjusting their synaptic strengths. Exposure to SR has been shown to alter the complexity of the dendritic tree in the HPC [12] and infralimbic cortex [13], alter synaptic functionality [7,9,19,20,21,22], produce LTP decrements in HPC synapses [23,24], and produce LTD decrements in the PFC [5]. Disruptions to each of these processes can adversely impact both excitatory and inhibitory neurotransmission [21]. We have previously shown that chronic, low dose rate neutron irradiation suppresses neuronal excitability and disrupts network level LTP within the HPC and mPFC [4]. While there is currently no information regarding HSP alterations following SR exposure, exposure to a 6 Gy dose of X-rays in juvenile, but not adult, Wistar rats resulted in both LTD and reduced LTP [25]. We have recently reported that LTD was reduced in the mPFC of rats exposed to ^28^Si ions [5].

The current study was designed to extend our understanding of the impact of chronic low dose and low dose rate neutron irradiation on HSP using two approaches. The first aspect of our study established low frequency stimulation (LFS) of LTD synaptic plasticity in the input L2-3 layer of the prelimbic (PrL) cortex of the mPFC in littermates of the C57Bl/6 mice used for LTP assessment [4]. The second part of our study used fluorescence assisted single synaptosome LTP (FASS-LTP) analysis to perform a cross-species validation of HSP (i.e., LTP) changes in the HPC of rats that were concurrently exposed to chronic low dose and dose rate neutron irradiation [16] with the C57Bl/6 mice [4]. We have previously shown that cognitive performance after SR exposure may arise due to compensatory changes in the composition of the HPC proteome, including differential expression of proteins involved in positive regulation of excitatory post-synaptic potential and protein localization to the synapse [26]. We thus determined whether ionotropic glutamatergic neurotransmission differed in neutron exposed rats that were able to encode SD associative recognition memory on the first attempt from those that were unable to do so.

Our data indicate that mPFC LTD in mice was reduced following neutron exposure, when our previous data on HPC LTD are considered, these data suggest a systemic change in the LTD of the hippocampal-cortical circuit. The FASS-LTP analysis of the hippocampi from rats identified a close to significant trend of reduced LTP after neutron exposure, but significant changes in the phosphorylation state of glutamate receptor subunits in rats that successfully encoded the SD associative recognition memory on the first attempt. Collectively these data suggest that SR exposure impacts HSP in the hippocampal-cortical circuit, and that the ability to successfully encode associative recognition memory is a dynamic, multicircuit process, possibly involving compensatory changes in AMPA receptor (AMPAR) density on the synaptic surface.

## 2. Results

### 2.1. Cortical Synaptic LTD Is Perturbed by Chronic Low Dose Neutron Irradiation in Mice

In previous studies of chronic effects of low dose neutron irradiation, there was reduced excitability of the hippocampal and mPFC neurons with suppressed frequency of excitatory synaptic inputs in irradiated mice [4]. In the current study, we extended the observations by studying the effects of such radiation exposures on LTD at the synapse by stimulating at layer 2/3 and recording of layer 5 of the mPFC. The specific reason to study this circuit is because layer 2/3 neurons receive sensory and motor inputs from other cortical areas, and layer 5 neurons generate local circuit outputs to subcortical structures that mediate planned behaviors [27].

Evoked extracellular fEPSPs reflect synchronous dendritic depolarization in neuronal population in vicinity of the recording electrode in response to presynaptic stimulus. These recordings are advantageous to detect diffuse neuronal or synaptic damage within the interrogated neuronal population. The magnitudes of the evoked fEPSPs tested at a range of stimulus intensities did not differ between sham-controls and irradiated mice indicating no radiation-induced decrements in neuronal cell number or excitatory synaptic circuits in the vicinity of the recording electrode (data not shown). There was also no difference in proportion of slices with fEPSP amplitude < 0.2 mV that could be ascribed to prior exposure to irradiation (data not shown).

Using paired-pulse and low-frequency stimulation (LFS) protocols, the impact that neutron exposure had on PPF and LTD (reflecting short-term and long-term synaptic plasticity, respectively) of fEPSPs was established. There were no significant differences between the sham and irradiated mice in the PPF (at 25, 50, 100 and 200 ms) in either L2-3 or L5 of the PrL cortex (Table 1), suggesting that there are no changes in the synaptic vesicle release or changes in the homeostatic levels. However, we did observe a significant change when the system was stressed using a low frequency stimulation dependent long-term depression protocol (LFS-LTD, Figure 1). After a 10-min baseline, when the slices were subjected to 15 min of a 1 Hz pulse, the field responses of the control animals registered a robust drop from the baseline levels (Figure 1A). However, LTD in the L5 was only weakly expressed in the cohort exposed to neutrons when compared to a relatively robust LTD in the sham controls (Figure 1B). This difference in LTD magnitudes was compared at 80–90 min reference time point and was confirmed to be statistically significant (Control: 50.28 ± 0.817 vs. Neutron: 68.62 ± 0.368, *n* = 14, **** *p* < 0.0001, unpaired non-parametric Mann–Whitney test). Thus, in addition to the LTP deficits reported in the earlier paper [4], the chronic irradiation of low dose neutron irradiation also causes LTD deficits in the mPFC synapses.

### 2.2. Decreased PSD-95 Expression and Increased CD68 Expression Is Observed in Chronic Low Dose Neutron-Irradiated Mice

We further explored the synaptic integrity of the hippocampal CA1 stratum radiatum by quantitatively assessing the number of PSD-95 and CD68 expression (Figure 2). Post-synaptic density protein 95 (PSD-95) is an excitatory-associated synaptic protein involved in plasticity, the level of which can be influenced by radiation exposure [11,28]. As PSD-95 is responsible for recruiting receptors and other proteins to the synaptic cleft, changes in expression level of PSD-95 may disrupt neurotransmission in a manner that could contribute to cognitive dysfunction. For that reason, we evaluated PSD-95 protein levels in the dentate gyrus (DG) region of the HPC (Figure 2A). Representative images clearly show neutron-induced decreases in PSD-95 staining 3.5 months after the end of neutron irradiation (Figure 2A). Quantitative analysis of the number of foci indicated a significant decrease in the DG region of the neutron-irradiated brain (Figure 2B; * *p* < 0.05).

Microglial activation has been used as a marker for neuroinflammation in multiple irradiation scenarios [29], and has been shown to be associated with space radiation-induced cognitive impairments [14] CD68 expression, a marker of microglial activation, was increased in the DG region of the HPC of the neutron-irradiated brain (Figure 2A). Quantification of CD68 immunoreactivity demonstrated a significant increase in CD68 at 3.5 months after the end of neutron exposure (Figure 2B; * *p* < 0.05).

### 2.3. Neutron Irradiation Results in Compensatory Increases in Apical Intersections and Dendritic Length, but Not Basal Dendritic Complexity

We assessed whether low dose neutron exposure had an effect on CA1 pyramidal neuronal dendritic length and branching complexity (Figure 3). Representative examples of CA1 dendritic arbor reconstructions are depicted in Figure 3A. Our results showed no significant differences in the length or both apical (Figure 3C) and basal (Figure 3F) dendrites between groups, although irradiated mice did trend to have an increase in length compared to controls (Figure 3E). We did see a significant increase in the branching complexity of neutron exposed mice compared to controls as indicated in the number of intersections (Figure 3D) and amount of dendritic material in the Sholl analysis (F_(6,60)_ = 3.0, * *p* = 0.01; F_(6,60)_ = 3.491, *** *p* = 0.005, respectively). Specifically, we found increases at 150 μm from the soma in apical intersections and at 150 μm and 180 μm from soma in dendritic length (Figure 3E). This increase in neutron exposed animals may be a means of compensation. We did not observe and changes in basal dendritic complexity (Figure 3F–H).

### 2.4. Chronic Low Dose Neutron Irradiation Results in Decreased Synapse Density and Results in Alterations in Myelination

Having found that neutron irradiation affects the complexity of apical dendrites of CA1 neurons at 150–180 mm from the soma, we subsequently examined synapse density morphology in the stratum radiatum, at the ultrastructural level using quantitative EM. Analysis of total synapse density did revealed significant decrease in synapse density between irradiated mice and controls (t_(7)_ = 3.610; ** *p* = 0.009; Figure 4A). When we looked at the densities of non-perforated and perforated synapses we found that irradiated mice had significantly fewer non-perforated synapses compared to controls (t_(7)_ = 4.457; *** *p* = 0.003). There was no difference in the density of perforated synapses (Figure 4B,C). To address whether neutron irradiation alters ultrastructural aspects of stratum radiatum synaptic morphology, we measured maximum head diameter, maximum PSD length and PSD area. We found no significant differences in mean PSD area, mead PSD length and mean head diameter (Figure 4D–F). Our previous studies have classified mouse CA1 spines with head diameters < 0.4 μm as thin spines and > 0.4 μm as mushroom spines [30,31,32,33,34]. When we apply this parameter to nonperforated synapses we found that size of the larger, >0.4 μm spines in neutron irradiated mice are smaller than in control mice suggesting that there may be fewer mushroom spines in irradiated mice compared to controls (Figure 4H, t_(285)_ = 2.061; * *p* = 0.04). Interestingly, larger nonperforated synapses from Neutron irradiated mice had significantly larger PSD lengths than sham mice, which may be some form of compensation for the loss of spines (Figure 4I; t_(404)_ = 3.725; *** *p* = 0.0002). There was no difference in perforated synapses (data not shown).

We next asked whether neutron irradiation had an effect on myelination. We quantified the percentage of myelinated axons as well as the g-ratio of axons (the ratio of inner-to-outer diameter of the myelinated axon) providing a morphometric analysis of the axons. We did not observe any significant differences in the percent of myelinated axons between neutron irradiated mice and controls (Figure 5C). Comparison of total g-ratios also did not show any differences between both groups of mice (Figure 5D). When we separated axons by size we found that the smaller caliber axons, <0.4 μm showed an increase in g-ratio indicative of thinner myelin sheaths, however this did not reach significance (Figure 5E, t(99) = 1.915; *p* = 0.058).

### 2.5. Chronic Low Dose Neutron Irradiation in Rats Modulates Glutamate Neurotransmission in Hippocampal Synapses

Collectively, these data indicate that neutron exposure induced HSP (LTP and LTD) changes in the mPFC and HPC of mice, accompanied by microglial activation and myelin changes. Clearly such changes suggest that neurological function may be impacted, but whether the HSP changes were associated with the neutron-induced loss of associative recognition memory was unclear. The neutron exposure experiments were devised to provide a cross-species (mouse/rat) validation of the impact of protracted (low dose rate) neutron exposures on cognitive flexibility [4,16]. The rat ATSET studies [16] established individual performance metrics (in each of the stages) and banked tissue from the individual rats. This bank of hippocampal samples from rats with defined associative recognition memory on SD task performance was an ideal opportunity to determine the functional significance of the observed LTP changes in the HPC after neutron exposure [4]. We thus establish LTP status in these archived rat hippocampal samples using the fluorescence assisted single synapse long-term potentiation (FASS-LTP) technique [35,36] and determined whether neutron exposure changed hippocampal LTP status in rats, and whether such changes were associated with associative recognition memory (SD task performance). The HPC samples contained the whole of the HPC (from a slice), and thus provided an assessment of LTP within the entire HPC and not just for a defined region as in previous studies. [4]. An additional advantage of using these “complete” HPC samples was the ability to establish changes in the expression of AMPAR in the same samples, possibly providing mechanistic insight into the underlying cause of the neutron induced alterations in HSP and/or associative recognition memory performance.

The status of synaptic functionality can be assessed by using flow-cytometry of isolated synaptosomes to study the integrity of the LTP signaling components such as GluR1 and Nrx1β, which are known to translocate at the synaptic membrane to facilitate the expression of LTP (see review by [37,38]). Synaptosomal size assessment was restricted to the rectangular area enclosed between 0.79–3.39 microns (see Figure 6A) that provided the range of synaptosomal particles that contain the essential synaptic components necessary for facilitating chemical potentiation [35,36,39]. Every experiment included appropriate controls to increase the reliability of the results and reduce the potential for false positives by including four controls (see Figure 6B)—no antibody, only primary, only secondary and cLTP in the presence of NMDA antagonist.

FASS-LTP analysis of the hippocampi from the sham cohort of rats (black bars, Figure 6C) revealed a potentiation of cLTP (122.8 ± 18.56, *n* = 11) compared to basal levels, while analysis of the hippocampi of the neutron-irradiated cohort (blue bars, Figure 6C) showed no potentiation in FASS-LTP compared to basal levels and in fact a small reduction in FASS-LTP (87.86 ± 10.76, *n* = 20). Calculation of the variance from basal levels for each individual rat, facilitated a quantitative assessment of FASS-LTP (Figure 6D). The hippocampi from the sham cohort were characterized as having a 29% increase in FASS-LTP over than that seen under basal conditions, whereas the entire neutron cohort of rats showed a small (6%) decrease in cLTP (Figure 6C). This differential of FASS-LTP levels between the cohorts just failed to reach significance (*p* = 0.059, ns) using the non-parametric one-tailed Mann–Whitney test. When the FASS-LTP data from the neutron exposed rats was stratified upon whether they passed the SD test (encoded associative recognition memory) or not, FASS-LTP levels in the rats that passed the SD stage were ~25% lower compared to basal levels (Figure 6D), which was significantly lower than in the shams (* *p* = 0.044, one-tailed Mann–Whitney test), but not from that seen in the rats that failed the SD stage.

### 2.6. Western Analyses of Radiation-Induced Changes in Synaptic Proteins

Western blot analysis of aliquots of the synaptosomal preparations used in the FASS-LTP analysis was thus commenced to establish whether the observed reduction of cLTP in the neutron-exposed rats, specifically those that had passed the SD stage, was attributable to changes in the composition of AMPA and NMDA receptor subunits. While there were no significant changes in the abundance of the GluA2 AMPA receptor subunit (Figure 7A) or of the GluN1 NMDA receptor subunit (Figure 7B), there was a significant decrease in the GluA1 subunit of the AMPAR complex in the neutron-exposed rats that had passed the SD task (blue filled bars, Figure 8A) compared to the other groups (sham, black filled bars; neutron irradiated and bad ATSET performance, open bars). Further investigation of the phosphorylation status of two serine residues that regulate GluA1 expression on the synaptic cleft revealed that while the phosphorylation status of serine 831 residue was unaffected in all three groups (Figure 8B), the serine 845 phosphorylation was significantly decreased in the neutron irradiated rats that had passed the SD stage (Figure 8C).

Collectively these data suggest that LTP within the HPC of rats exposed to neutrons is reduced relative to that observed in sham rats, and that neutron exposed rats that had passed the SD task by encoding associative recognition memory exhibited more pronounced reductions in cLTP than in those that failed to pass the SD task. There was a high degree of similarity between the observed changes in FASS-LTP and the phosphorylation of the GluA1 serine 845 residues.

## 3. Discussion

Problem solving is a fundamentally important skill that enables individuals to adapt to new situations. The encoding and retrieval of the memory of previously experienced situations are critical for the future resolution of the same situation, or via transitive inference the resolution of problems that require similar solutions to the previously encountered situation. The mechanistic basis for learning and memory is complex but it is clear that HSP plays a key role in encoding such memories. Our previous studies have revealed that mice and rats exposed to protracted neutron exposure exhibit behavioral performance decrements consistent with defects in the encoding or retrieval of associative recognition memory, a process that is highly dependent upon two hippocampal-cortical circuits, involving the HPC, PRH, and mPFC cortices [17]. The current study has sought to expand our knowledge of how SR impacts HSP within the HPC axis.

Local circuits within and between layers of cortex play an important role in receiving, converging, integrating, and conveying the physiological signals that are essential for higher order executive functions. The coordinated output from the infragranular layer 5 cells that receive convergent excitatory inputs from layer 2/3 pyramidal cells of the mPFC are purported to regulate information to other cortical and subcortical targets in performing executive functions [27]. Our data (Figure 1) demonstrate that chronic neutron exposure leads to a persistent (at least 4 months) reduction in LTD within layer 5 cells of the mPFC of mice, in addition to our previously reported reduction of LTP in the same cells [4]. The LTD studies reported here were conducted in a separate, but contemporaneous cohort of mice to those used in the LTP studies, thus it is impossible to definitively state that both LTD and LTP would have been reduced in individual mice following neutron exposure, but at the cohort level there were significant reductions in both LTP and LTD.

The classic concept of a bi-directional regulation of HSP has a push-pull relationship between LTP and LTD, but after neutron exposure this relationship appears to have been permanently altered. Whether this is a systemic response to SR is not known. While there have been several studies that have reported SR induced changes in LTP, there has been just one study reporting that exposure to 10 cGy of Si ions resulted in reduced LTD, also within layer 5 cells of the mPFC of rats [5]. Following X-ray exposure of juvenile, but not adult rats the classic push-pull balance between LTP and LTD does seem to be preserved [25].

Further investigation of the causes of the concomitant loss of LTP and LTD in the mPFC of mice are clearly required. Reductions in LTP and LTD could reflect a neutron-induced change in intrinsic synaptic membrane properties similar to the decreased neuronal excitability in CA1 neurons after proton exposure [22]. However, we did not observe any overt changes in the input-output ratio (see Figure 9) in the present study, nor in our previous study on LTD changes in the rat mPFC after ^28^Si irradiation [5]. Moreover, there were no detectable differences in the intrinsic properties of presynaptic membranes (Table 1 and Figure 10). To date, our data suggest that the reduced LTP/LTD following neutron exposure may arise as a result of post-synaptic changes. The loss of both LTP and LTD following neutron exposure may be attributable to the loss of PSD95. PSD-95 is a key excitatory-associated synaptic protein responsible for recruiting receptors and other proteins to the synaptic cleft.

Neutron exposure could also influence LTP/LTD by impacting one of the multiple signaling cascades that regulate neuromodulator receptor functionality. The processes regulating HSP are complex and very context specific. In the visual cortex spike-timing-dependent plasticity is shaped by neuromodulator receptors coupled to adenylyl cyclase (AC) and phospholipase C (PLC) signaling cascades [40]. Activation of the AC and PLC cascades results in phosphorylation of postsynaptic glutamate receptors at sites that serve as specific “tags” for LTP and LTD. As a consequence, the outcome (i.e., whether LTP or LTD) of a given pattern of pre- and postsynaptic firing depends not only on the order of the timing, but also on the relative activation of neuromodulator receptors coupled to AC and PLC [40]. Simultaneous reductions in both LTP and LTD have been previously observed in mice that bear a mutation at the S845 residue on GluR1 [40].

Synaptic plasticity is also influenced by multiple factors within the neuron, including what cell the neuron is interacting with even along the same axon [41,42]. Furthermore, the observed changes in HSP following neutron exposure may be the consequence of neutrons impairing the function of the cells that maintain neuronal function (e.g., astrocytes, oligodendrocytes, and microglia). Oligodendrocytes are essential for providing metabolic support to neurons, rapidly transferring short-carbon-chain energy metabolites like pyruvate and lactate to neurons through cytoplasmic “myelinic” channels and monocarboxylate transporters [43]. Astrocytes are in direct contact with their neuronal pre- and postsynaptic counterparts and release soluble factors to modulate synaptic transmission of both excitatory and inhibitory synapses. Microglia interact closely with neurons, oligodendrocytes, and astrocytes. Imaging studies have shown that microglia extend and retract their processes continuously to survey their local environment in the healthy brain [44,45]. Upon sensing CNS injury, microglia are activated, secreting both proinflammatory mediators, such as tumor-necrosis-factor-(TNF-) α [46,47] or interleukin-(IL-) 1β, nitric oxide (NO) [48,49], and glutamate [50], along with anti-inflammatory effectors, such as IL-4 and IL-13, which can enhance neuronal survival [51,52]. Depending on the predominance of factors secreted, microglia can delineate into proinflammatory (M1) or anti-inflammatory (M2) phenotypes [53].

Our initial efforts to elucidate the mechanistic basis for the reduced LTD/LTP in neutron exposed mice revealed that in addition to the previously reported changes in the dendritic tree structure there was marked increase in CD68+ microglia. The increased presence of activated microglia, contributing to the prolonged proinflammatory environment, could underlie the changes in both LTP and LTD seen in the neutron exposed mice. Microglial activation can alter the AMPAR/NMDAR ratio and the ratio of AMPAR-over NMDAR-mediated currents [54]. There are multiple signaling interactions between the microglia, astrocytes and neurons that could be responsible for such changes. The microglia/astrocyte mediated release of the cytokine, TNF-α, alters the surface expression of AMPA receptors in neuronal cultures, and alters synaptic strength [55]. Minocycline induced blockage of microglial activation induces LTD in the C fibers of the dorsal horn of the spinal cord involving Src family kinases and TNF-α [56]. Furthermore, reductions and structural changes to the state of myelin in the irradiated brain would have obvious ramifications for neurotransmission, and past work has indicated that single ion exposures caused a significant increase in the percentage of unmyelinated axons in the hilar region of the HPC [30]. In the present study, while we did not observe any significant changes in the myelin thickness (except for a trend in the smaller axons, Figure 5), we did observe a decrease in the total synaptic density that is attributed to the decrease in the non-perforated synapses (but not perforated synapses) in neutron irradiated mice compared to sham (Figure 4). Loss of non-perforated spines could affect conductance scaling and protein sequestration [57] negatively because they make up 80% of the total axospinous synapses [58]. Moreover, there are reports suggesting that 40% of all non-perforated synapses lack AMPARs and make up silent synapses that are important for plasticity [57] and this indicates that the loss of AMPARs that we observe in the neutron-irradiated group (Figure 8C) corroborates with a potential mechanism involving decreased AMPAR response that affects non-perforated spines and underlying functional deficits seen in behavioral outcomes reported earlier [4,16]. We also report here that there are bigger PSDs associated with the non-perforated synapses in the neutron group (Figure 4H), but the overall PSD-95 foci are reduced (Figure 2B), perhaps a futile attempt by the synaptic homeostasis mechanisms to dynamically compensate effects of neutron irradiation. In an earlier study by our group using high energy charged particles (^16^O, ^28^Si, ^4^He), we have reported similar changes in the perforated vs. non-perforated synapses [30], but unlike our present observations with neutron irradiation (Figure 5C), we did observe decreased myelination. We speculate that such a distinction in myelination effects could be attributed to the absence of secondary delta radiation tracks that the protracted photon contamination in the current experiments could not elicit, making this an interesting topic for future investigations.

Collectively, these data indicate that neutron exposure induced HSP (LTP and LTD) changes in the mPFC and HPC of mice, accompanied by microglial activation and altered myelination. The present study was part of a larger study focused on a cross-species (mouse/rat) validation of the impact of protracted, low dose rate neutron exposures on cognitive flexibility [4,16]. Using FASS-LTP analysis we observed an almost (*p* = 0.059) significant reduction in LTP of hippocampal synapses from rats exposed to neutrons (Figure 6C). These data are consistent with our prior report of reduced hippocampal LTP (fEPSPs) in mice exposed to chronic neutron radiation [4]. The fact that the same neutron exposure paradigm caused similar reductions in hippocampal LTP in both rats and mice using two very different methodological approaches (i.e., fEPSP in the CA1 region and FASS-LTP in synaptic preparation from the entire HPC) suggests that neutron-induced suppression of LTP is a bona fide effect.

ATSET studies in rats [16] have established individual performance metrics in each of the stages of testing, thereby providing an opportunity to determine if LTP varied in irradiated rats that passed or failed to encode associative recognition memory in the SD task. As observed in our previous studies, there are marked individual differences in the response of the rat brain to SR exposure, whether at the behavioral [5,6,7,8,9,10] or cellular level protein [26]. At the cohort level, neutron exposed rats showed little to no variation in LTP compared to baseline, however at the individual level, roughly half the animals showed increased LTP while the other half showed decreased LTP. Interestingly, the majority of the rats that passed the SD task showed decreased LTP, whereas there was an equal number of rats with elevated and reduced LTP for those that failed the task.

The FASS-LTP assays measure the insertion of AMPAR into the post-synaptic membrane, (in this study GluR1 and Nrx1ß) after chemical stimulation (KCl and glycine). Despite rigorous control validation, cLTP may be incomplete, suggesting the removal and/or displacement of GluR from the synaptic membrane. Whether this results from loosely bound GluR1 or competitive/non-competitive insertion of other synaptosomal receptors is uncertain. Western blot analyses on ionotropic glutamate receptors, known to affect LTP and LTD, in aliquots of the same rat hippocampal synapse preparations used for FASS-LTP, revealed no major differences in the expression levels of GluR1 receptor subunits. However, there was a differential decrease in cLTP (Figure 6D) and the phosphorylation of serine at 845 of the GluA1 subunit in the irradiated rats showing good ATSET (Figure 8C). Reasons for this are uncertain, but past work has shown that mutation of the S845 residue on the GluR1 that blocks phosphorylation reduced both LTP and LTD [40].

The role of hippocampal LTP in associative learning is extremely complex [59] and can differ substantial from classical conditioning. Hippocampal synapses are modified in strength during the acquisition of classical conditioning but not instrumental, learning tasks [59]. How the present data fits within the context of associative learning paradigms remains uncertain, but the cross-species reductions in mPFC LTD following neutron exposure suggest, that network level changes in hippocampal-cortical circuitry are a target for certain SR-induced neurocognitive deficits. A further extension of this work looking at the effect of stress and glucocorticoids on the HPC will be important for the Hippocampus-Prefrontal Cortex connectivity [60]. Since corticosterone levels can alter levels of AMPARs due to secondary effects associated with PSD-95/Stargazin fixation and the extrasynaptic mobility that occurs [61], measurement of corticosterone levels will be considered in future experiments as an additional measure of stress influencing the hippocampal-prefrontal cortex connectivity and neuromodulation influenced by space radiation. It will also be relevant to look at the amygdala related changes in this new light since the plasticity in the amygdala is known to occur in the opposite direction in neuropsychiatric disorders [62]. Incorporating such questions that emerge from research in the neuropsychiatric field will be instrumental in advancing our efforts to enhance the understanding of long-term effects of space radiation. Moreover, an aspect of research that spawns from our observation, but has not been effectively addressed in space radiation studies, is the need for assessing how long-term isolation affects synaptic function affecting decision-making under stress, which is a concern that is known in airline pilots [63]. Collectively these data suggest that the impact of SR exposure on HSP cannot be defined at the cellular level, but rather, is reliant on multiple interacting circuits that can be recruited in a context-specific manner that affect the ability to successfully encoded associative recognition memory.

## 4. Materials and Methods

### 4.1. Animal Welfare

This study was conducted in accordance with the National Research Council’s “Guide for the Care and Use of Laboratory Animals (8th Edition)” at the animal care facilities of Colorado State University (CSU), Eastern Virginia Medical School (EVMS), The University of Texas Medical Branch at Galveston (UTMB) and University of California, Irvine (UCI), all of which are accredited by the Association for Assessment and Accreditation of Laboratory Animal Care, International. All procedures were approved by the Institutional Animal Care and Use Committees of all institutions.

### 4.2. Rodent Subjects

Six-month-old male (*n* = 73 total) C57BL/6 mice were delivered to CSU direct from the suppliers (Jackson Laboratory; Bar Harbor, ME), where they group housed, maintained on a standard 12:12 light/dark cycle, and given ad libitum food and water. Following acclimation at CSU, the rodents were randomly assigned to either the sham or neutron exposure cohorts. The mice were subjected to a prolonged six-month (180 day) exposure at the CSU neutron irradiation facility, with a total dose of 18 cGy being delivered [4,16]. Details of the physical properties of the CSU facility and characterization of the radiation spectrum and dosimetry are outlined in Borak et al. (2019). The mice were exposed to neutrons for 15.9–18.3 h/day (accounting for the decay of the source), after which daily animal husbandry was performed. The sham control rodents were similarly housed in an adjacent facility. One week after the end of the exposure period the mice were divided into two cohorts, one being shipped to UCI for OIP performance evaluation, immunohistochemistry and processing for ultrafine structure analyses (*n* = 23 sham and 22 irradiated, respectively) [4], and the other being shipped to UTMB for LTD assessment (n = 14 shams and 14 irradiated). At each institute the mice were group housed, maintained on a standard 12:12 light/dark cycle, and given ad libitum food and water. Approximately 90 days from the end of neutron exposure, cognitive performance and electrophysiology measurements were performed on irradiated and sham mice, now aged between 17–18 months.

Contemporaneously with the mouse neutron exposure, male Wistar rats (Hla^®^(WI)CVF^®^; Hilltop Lab animals, Inc., Scottsdale, PA, USA) that had been pre-selected for a high degree of performance in the ATSET test were shipped to CSU and exposed in the same facility (and thus the same manner) as the mice. After completion of the neutron exposure the rats (n = 9 shams and 27 irradiated, respectively) were shipped back to EVMS, where they were maintained on a reversed 12:12 light/dark cycle and given *ad libitum* access to Teklad 2014 (Envigo, Indianapolis, IN) chow and municipal water by bottle. Approximately 90 days from the end of neutron exposure the rats were assessed for ATSET (cognitive flexibility) performance [16] (~21 months of age at the time of testing). One week after the completion of the cognitive flexibility testing the HPC of the rats were recovered. To avoid inducing changes in the proteome of the HPC due to anesthesia or asphyxiation, the rats were euthanized by guillotine. The brain was immediately recovered, and the HPC recovered in accordance with our previous protocol [19]. The excised HPC was placed in a sterile 1.5 mL Eppendorf tube, and then immersed into liquid nitrogen. The snap-frozen hippocampi were stored at −80 °C until being shipped to UTMB for FASS-LTP analysis. HPC samples from 7 sham and 21 neutron exposed rats were shipped to UTMB coded to prevent experimenter bias. Rats were classified as either passing the SD stage on the first attempt (SD-pass) indicating these rats were able to SD associative recognition memory, of they failed to complete the SD stage (SD-fail).

### 4.3. Field Electrophysiological Recordings

Our standard protocol was used as previously described [64,65,66,67,68,69,70,71] with the following modifications for recording low frequency stimulation long-term depression (LFS-LTD). Briefly, mice were deeply anesthetized with isoflurane (Piramal Pharma Solutions, Lexington, KY, USA) and transcardially perfused with ~30 mL of room temperature carbogenated (95% O_2_ and 5% CO_2_ gas mixture) NMDG-artificial cerebrospinal fluid (aCSF) (in mM 93 *N*-Methyl-d-Gluconate, 2.5 KCl, 1.2 NaH_2_PO_4_, 30 NaHCO_3_, 20 C_8_H_18_N_2_O_4_S, 25 C_6_H_12_O_6_, 5 C_6_H_7_O_6_Na, 2 CH_4_N_2_S, 3 C_3_H_3_NaO_3_, 10 MgSO_4_,7H_2_O, 0.5 CaCl_2_,2H_2_O, 12 C_5_H_9_NO_3_S, pH 7.4). The brains were removed and sliced using Compresstome VF-300 (Precisionary Instruments, Greenville, NC) in carbogenated NMDG-aCSF to obtain 350 μm coronal sections. Slices were allowed to recover for 10 min in carbogenated NMDG-aCSF at 33 °C. Slices were then maintained at room temperature in a modified carbogenated HEPES holding aCSF solution (in mM 92 NaCl, 2.5 KCl, 1.2 NaH_2_PO_4_, 30 NaHCO_3_, 20 C_8_H_18_N_2_O_4_S, 25 C_6_H_12_O_6_, 5 C_6_H_7_O_6_ Na, 2 CH_4_N_2_S, 3 C_3_H_3_NaO_3_, 2 MgSO_4_,7H_2_O, 2 CaCl_2_, 12 C_5_H_9_NO_3_S, pH 7.4). Slices were recorded in carbogenated standard recording naCSF (in mM 124 NaCl, 2.5 KCl, 1.2 NaH_2_PO_4_, 24 NaHCO_3_, 5 C_8_H_18_N_2_O_4_S, 13 C_6_H_12_O_6_, 2 MgSO_4_, 7H_2_O, 2 CaCl_2_, pH 7.4). Evoked field excitatory post-synaptic potentials (fEPSPs) recordings were performed by stimulating Layer 2/3 using a stimulating electrode of ~22 kΩ resistance and glass recording electrodes in the Layer 5 region. Current stimulation was delivered through a digital stimulus isolation amplifier (A.M.P.I, Israel) and set to elicit a fEPSP approximately 30% of maximum for synaptic potentiation experiments using platinum-iridium tipped concentric bipolar stimulating electrodes (FHC Inc., Bowdoin, ME, USA). The use of platinum iridium wire and diphasic pulses can help minimize electrode polarization. Using a horizontal P-97 Flaming/Brown Micropipette puller (Sutter Instruments, Novato, CA, USA), borosilicate glass capillaries were used to pull recording electrodes and filled with naCSF to get a resistance of 1–2 MΩ. Field potentials were recorded in CA1 stratum radiatum using an Ag/AgCl wire in CV7B headstage (Molecular Devices, Sunnyvale, CA, USA) located ~1–2 mm from the stimulating electrode. LTD was induced using a low frequency stimulation protocol (900 × 1 Hz). Recordings were digitized with Digidata 1550B (Molecular Devices, Sunnyvale, CA), amplified 100X and digitized at 6 kHz using an Axon MultiClamp 700B differential amplifier (Molecular Devices) and analyzed using Clampex 10.7 software (Molecular Devices). To assess basal synaptic strength, 250 μs stimulus pulses were given at 10 intensity levels (range, 100–1000 μA) at a rate of 0.1 Hz. Three field potentials at each level were averaged, and measurements of fiber volley (FV) amplitude (in millivolts) and fEPSP slope (millivolts per millisecond) were performed using Clampfit 10.7 software. Synaptic strength curves were constructed by plotting fEPSP slope values against FV amplitudes for each stimulus level. Baseline recordings were obtained for 10 min by delivering single pulse stimulations at 20 s intervals. All data are represented as a percentage change from the initial average baseline fEPSP slope obtained for the 10 min prior to LFS. Two slices were recorded per animals and were averaged to provide the data as number of animals per group.

### 4.4. Fluorescence Assisted Single Synaptosome Long Term Potentiation (FASS-LTP)

Modifying critical steps of two published articles [35,36], we adapted and recently published [72] a chemically induced long-term potentiation technique (cLTP) utilizing synaptosomal induction by glycine and tracking insertion of glutamate AMPA receptors (GluR1) into the postsynaptic surface. These studies reported using freshly prepared synaptosomes immediately following euthanasia in mice or death in human patients. We optimized the protocol to allow for the assessment of synaptosomal FASS-LTP in any frozen brain tissue regardless of post-mortem interval. Hippocampal synaptosome P2 fractions were obtained by Syn-PER extraction method as described previously [73]. Briefly, high potassium chloride concentration depolarizes synaptosomes releasing endogenous glutamate, which activates synaptic NMDA receptors in conjunction with the NMDAR co-agonist glycine. Both glutamate and glycine, then, open receptor channels, facilitating calcium influx into synaptosomes, which, in turn, initiates a cascade of events resulting in the translocation of AMPARs from internal pools into synaptic sites. The synaptosomes are then incubated with antibodies specific for extracellular epitopes on GluR1 and neurexin-1β (Nrx1β), a presynaptic adhesion molecule stabilized at the membrane surface by synaptic activity. The GluR1^+^Nrx1β^+^ double labeling ensures that intact synaptosomes contain both presynaptic and post-synaptic elements. Solutions for FASS-LTP and procedures followed the protocol published previously [35,36] with the following exception—P2 fractions containing 5,000,000 synaptosomes were suspended in 200 μL extracellular solution, one each for extracellular (E), basal (B) and 200 μL of cLTP (C) solution containing tubes. Extracellular solution contains (in mM): 120 NaCl, 3 KCl, 2 CaCl_2_, 2 MgCl_2_, 15 glucose, and 15 HEPES, pH 7.4; whereas cLTP solution is Mg^2+^-free and contains (in mM): 150 NaCl, 2 CaCl_2_, 5 KCl, 10 HEPES, and 30 glucose, pH 7.4. A 30-min incubation at room temperature (RT) on a slowly oscillating shaker was done to recover the prep from the frozen state to respond to the biochemical steps. This critical step was essential before adding 0.001 mM strychnine and 0.02 mM bicuculline methiodide in tube C alone for activation of the receptors. Equivalent amounts of extracellular solution were added to control tubes E and B. For stimulation, 20 μL of 5 mM glycine was added only to tube C whereas equal volume of extracellular solution was added to tubes E and B. Following stimulation, synaptosomes in tube C was depolarized with 100 μL of solution containing (in mM) 50 NaCl, 100 KCl, 2 CaCl_2_, 30 glucose, 10 HEPES, 0.5 glycine, 0.001 strychnine, 0.02 bicuculline and incubated at 37 °C for 30 min. 100 μL of extracellular solution was added to tubes E and B and incubated along with tube C for 30 min at 37 °C. Following the incubation, contents in tubes E, B and C were transferred to 15 mL centrifuge tubes. In a sequential step, ice-cold solutions of 0.5 mL of 0.1 mM EDTA-PBS and 4 mL of 5% blocking buffer (fetal bovine serum in PBS) were added to all the tubes E, B and C to stop the reaction. Tubes were chilled on ice and centrifuged at 2500 g for 5 min at 4 °C. Supernatant was discarded and 2.5 μg/mL primary antibodies—GluR1 (Rabbit polyclonal ABN241, EMD Millipore, Burlington, MA, USA) and Nrx1β (mouse monoclonal N170A/1, Antibodies Inc., Davis, CA, USA)—in blocking buffer were added only to pellets of tubes B and C. To pellet E, 100 μL of blocking buffer was added. After incubation on ice with agitation, tubes E, B and C were washed twice with 1x PBS at 2500 g for 5 min. Pellets were resuspended in 2.5 μL/mL of secondary antibody (anti-rabbit Alexafluor 488 and anti-mouse Alexafluor 647, Invitrogen, Carlsbad, CA, USA) in 400 μL were added. After 30 min of incubation at 37 °C for 45 min protected from light, synaptosomes were washed twice at 2500 g for 5 min on each wash. Supernatant was discarded and at the least 1 mL of solution of pellets were left undisturbed. 400 μL of 2% paraformaldehyde in PBS was used to resuspend the pellets. Samples were subjected to flow cytometry using Guava easycyte™ 8 flow cytometer (GuavaSoft 2.7 Software). Relative size and granularity were determined by forward scatter (FSC) and side scatter (SSC) properties (see Figure 6A). FSC, SSC, and fluorescence [Red2 fluorescence (642 nm) and Green (488–532 nm)] signals were collected by using log amplification. Identical FSC settings were used for acquiring data on bead standards and samples. Synaptosome integrity were routinely assessed using standard electron microscopy [73]. Two-color parameter density plots showing Nrx1β and GluR1 surface detection in size-gated synaptosomes before and after cLTP. GluR1–Nrx1β double-positive events (upper right quadrant) increase after cLTP. For each experimental sample, the ratio of cLTP (C)/basal (B) x 100 provided the percent association of Nrx1β-GluR1 providing a quantifiable measure of the cellular potentiation strength. The greater the ratio, the better the synaptic integrity of the system. Depending upon the availability of each of these tissue types, one to three technical repeats were performed, and the values averaged to obtain mean and standard error of mean. To accommodate for the sample numbers, non-parametric one-way ANOVA (Kruskal–Wallis) followed by Dunn’s post hoc was used for statistical significance at *p* < 0.05.

### 4.5. Western Blot Analysis

The bicinchoninic acid (Pierce BCA, Thermofisher, Waltham, MA, USA) assay method was used for protein estimation to prepare samples of equal protein concentration. Samples were prepared in 4 x Sample buffer containing DTT (Invitrogen, Carlsbad, CA, USA) and boiled for 5 min prior to loading. Forty micrograms of protein were loaded with appropriate marker on 10% SDS-PAGE gels. Western Blot was performed using a vertical electrophoretic chamber (Bio-Rad, Hercules, CA, USA) set at 100 V for the running and at 400 mA for the transfer of samples into a nitrocellulose membrane (0.22 μm pore size, Amersham, UK). The membrane was blocked using Odyssey blocking buffer (LI-COR, Lincoln, NE, USA) for 1 h at room temperature. Primary antibodies (see list below) were diluted 1:1000 in 1X TBST and incubated with the membrane at 4 °C overnight and 1 h at room temperature for β-actin (1:5000 dilution of Sigma Aldrich, St. Louis, MO mouse monoclonal antibody cat# A2228). The membrane was washed thrice with 1X TBST for 10 min each and incubated with LI-COR secondary antibodies diluted at 1:10,000 in Odyssey blocking buffer for 1 h at RT. The membrane was again washed thrice for 10 min each. Antibodies used in this study include: GluN1 (Cell Signaling Technology, Danvers. MA Cat#5704), GluA2 (Cell Signaling Technology, Danvers. MA Cat#13607), GluA1 (Cell Signaling Technology, Danvers. MA Cat#13185), phospho GluA1 @ Serine 831 (Cell Signaling Technology, Danvers. MA Cat#75574) and phosphor GluA1 @ S845 (Cell Signaling Technology, Danvers. MA Cat#8084). Western blots were imaged using LI-COR Odyssey infrared imaging system (LI-COR, Lincoln, NE, USA), application software version 3.0.30. The density of each immunoreactive band was measured using Image J (https://imagej.nih.gov/ij/ (accessed on 11 March 2021)) software.

### 4.6. Immunohistochemistry, Confocal Microscopy, Image Processing and 3D Quantification

Mice were deeply anesthetized using isoflurane and euthanized via intracardiac perfusion using 4% paraformaldehyde (PFA; Sigma Aldrich, St. Louis, MO, USA) in 100 mM phosphate buffered saline (PBS; pH 7.4, Gibco, Thermofisher, Waltham, MA, USA). Brains were cryoprotected (10–30% sucrose gradient) and sectioned coronally into 30 μm using a cryostat (Leica Microsystems, Wetzlar, Germany). For each endpoint 2 representative coronal brain sections from each of 4 animals per experimental group were selected at approximately 15 section intervals and stored in tris buffered saline (TBS, 100 mM, pH 7.4, Sigma-Aldrich, St. Louis, MO, USA). For the immunofluorescence labeling of PSD-95 and microglial activation marker CD68 mouse anti–PSD-95 (Thermo Scientific, Waltham, MA, USA; 1:1000) primary antibodies were used with Alexa Fluor 594 secondary antibody (1:1000). Tissues were then DAPI nuclear counterstained and sealed in slow fade/antifade mounting medium (Life Technologies, Carlsbad, CA, USA).

The immunostained coronal brain sections were scanned using a confocal microscope (Nikon Eclipse Ti C2) equipped with a 40 X PlanApo oil-immersion lens (1.3 NA, Nikon) and an NIS-Elements AR interface (v4.30, Nikon). 30 z stacks (1024-bit depth) at 0.5 µm from three different fields (318 × 318 × 24 µm^3^) in each section were imaged from the dentate gyrus and/or from the CA1 subfields. The digitized z stacks were deconvoluted using the AutoQuant software (version X3.0.4, Media Cybernetics, Rockville, MD, USA). An adaptive, 3D blinded method was used to create deconvoluted images for direct import into the Imaris module (version 8.1.2, Bitplane, Inc., Zurich, Switzerland). The 3D algorithm-based surface rendering and quantification of fluorescence intensity for each fluorescently labeled marker was carried out in Imaris at 100% rendering quality. Each channel was analyzed separately. 3D surface rendering detects immunostained puncta or nuclear staining (DAPI) satisfying pre-defined criteria, for the puncta size (0.5 to 1 µm) verified visually for accuracy. Using deconvoluted confocal z stack volume from the control group (untreated, unirradiated) as a baseline for the minimum thresholding, a channel mean intensity filter was applied and used for all the experimental groups for each batch of molecular markers. The pre-set parameters were kept constant throughout the subsequent analysis of immunoreactivity for each antigen. To maintain uniformity among the varying number of puncta for each individual time point and/or antigen analyzed, the number of puncta per 318 × 318 × 24 µm^3^ was normalized to control and data was expressed as a mean immunoreactivity (percentage) relative to unirradiated controls.

Post-mortem mouse tissue was also sent to the Icahn School of Medicine at Mount Sinai and the Uniformed Services University of Health Sciences (USUHS) for the morphological analysis of CA1 pyramidal neurons. Mice were deeply anesthetized using isoflurane and euthanized via intracardiac perfusion using 1% PFA in PBS (pH 7.4) followed by 4% PFA/0.125% glutaraldehyde in PBS for 12 min, as described previously [31,32,74]. The brains were removed and postfixed overnight at 4 °C in 4% PFA/0.125% glutaraldehyde in PBS. The brains were then sectioned on a Vibratome (Leica VT1000S, Leica Biosystems, Nussloch, Germany) at 200 μm for intracellular dye injections and at 250 μm thickness for electron microscopy (EM). All sections were stored at 4 °C in PBS  +  0.01% sodium azide until ready for use.

### 4.7. Intracellular Dye Injections

For intracellular injections, sections were incubated in DAPI (Vector Labs) for 5 min to reveal the cytoarchitectural features of the pyramidal layer of CA1 of the HPC. The sections were then mounted on nitrocellulose paper and immersed in ice-cold 0.1 M PBS. Pyramidal neurons in the CA1 were subjected to an intracellular iontophoretic injection of 5% Lucifer Yellow (Invitrogen) in distilled water under a direct current of 3–8 nA until dye had completely filled distal processes [33,34,75]. Five to 10 neurons were injected per slice and placed far enough apart to avoid overlapping of their dendritic trees. Brain sections were then mounted on gelatin-coated glass slides and cover slipped in Fluoromount G slide-mounting media (Southern Biotech, Birmingham, AL, USA).

### 4.8. Neuronal and Dendritic Reconstruction

Neurons were manually traced and reconstructed in 3-dimensions with a 63×/1.4 N.A., Plan-Apochromat oil immersion objective on a Zeiss Axio Imager Vario microscope equipped with a motorized stage, video camera system, and Neurolucida morphometry software (MBF Bioscience, Williston, VT, USA). To be included in the analysis, a loaded neuron had to satisfy the following criteria: (1) reside within the pyramidal layer of the CA1 as defined by cytoarchitectural characteristics; (2) demonstrate complete filling of dendritic tree, as evidenced by well-defined endings; and (3) demonstrate intact tertiary branches, with the exception of branches that extended beyond 50 μm in radial distance from the cell soma [33,34,75].Using NeuroExplorer software (MBF Bioscience) total dendritic length, number of intersections, and the amount of dendritic material per radial distance from the soma, in 30-μm increments [34]. were analyzed in order to assess morphological cellular diversity and potential differences between the animal groups. A total of 50 cells were reconstructed for controls (~8 cells per animal) and a total of 49 cells were reconstructed for neutron irradiated mice (~8 cells per animal).

### 4.9. Quantitative Analysis of Synapses and Myelin

Coronal sections (250 μm-thick) encompassing the CA1 region of the HPC were prepared for EM imaging as reported previously [30,31,32,33]. Once the tissue was processed, block faces were trimmed, and ultrathin sections (80 nm) were cut with a diamond knife (Diatome, Hatfield, PA, USA) on an ultramicrotome (Reichert-Jung) and at least 5 serial sections were collected on formvar/carbon-coated nickel slot grids (Electron Microscopy Sciences, Hatfield, PA, USA). All tissue samples were imaged on a Hitachi H-7500 transmission electron microscope using an NANOSPRT camera (Advanced Microscopy Techniques, Woburn, MA, USA). For synapse analysis, serial sections located in the SR of the hippocampal CA1 field were imaged at 2500×. An unbiased stereological approach using the physical dissector was performed to measure CA1 SR synapse density and analyzed in Adobe Photoshop (version CC 2018 19.1.2, Adobe Systems, San Jose, CA, USA), as described in our previous work [30,31,32,33]. EM tissue preparation and imaging was performed at The Microscopy CoRE and Advanced Bioimaging Center at the ISMMS. Axospinous synapse density was calculated as the total number of unique counted synapses divided by the total volume of the dissector (area × height of dissector). The criteria for inclusion as an axospinous synapse included the presence of a presynaptic terminal and a distinct PSD separated by a clear synaptic cleft. The same volume was sampled for each group. In addition to total synapse density, we also measured the densities of nonperforated and perforated synapses which were defined by the presence of a discontinuity in the PSD. A single person, blinded to each of the treatment groups, performed all analyses.

To characterize the degree of myelination, we calculated the percent of myelinated axons from 12 randomly selected, nonoverlapping fields of the hippocampal sulcus from each animal imaged at 4000×. An additional six randomly selected, nonoverlapping images were taken per animal at 7000× to evaluate myelin sheath thickness through g-ratio analysis. To calculate the g-ratio, the average diameter for each axon was divided by the average axon diameter plus twice the average myelin width [30,76,77]. Myelin regions that exhibited fixation artifacts or noncompaction were excluded from the analysis.

### 4.10. Statistics

All data are reported as mean ± SEM. Statistical significance was calculated using GraphPad Prism 8 (San Diego, CA, USA). All statistical tests were 2-tailed, with the threshold for statistical significance set at 0.05. To account for non-normal distribution of data, either non-parametric *t*-tests (Mann–Whitney U or Wilcoxon rank sum) or one-way ANOVA (Kruskal–Wallis test) followed by Dunn’s multiple comparison when significance was achieved, were used. Immunohistochemistry data are presented as Mean ± SEM where *n* = 2 slices from each of 4 mice/group. *p* values were derived from unpaired Student’s *t* tests. * *p* < 0.05. For neuronal structure analyses, *p* values for length were derived from unpaired Student’s *t* tests and two-way repeated measures for Sholl analyses. For EM analysis of synapses and myelination, *p* values were derived from unpaired Student’s *t* tests. The α level was set at 0.05 with values of *p* < 0.05 considered statistically significant. All data were represented as mean  ±  SEM. All statistical analyses were carried out using Prism software (GraphPad Software, San Diego, CA, USA).

## Figures and Tables

**Figure 1 ijms-22-03668-f001:**
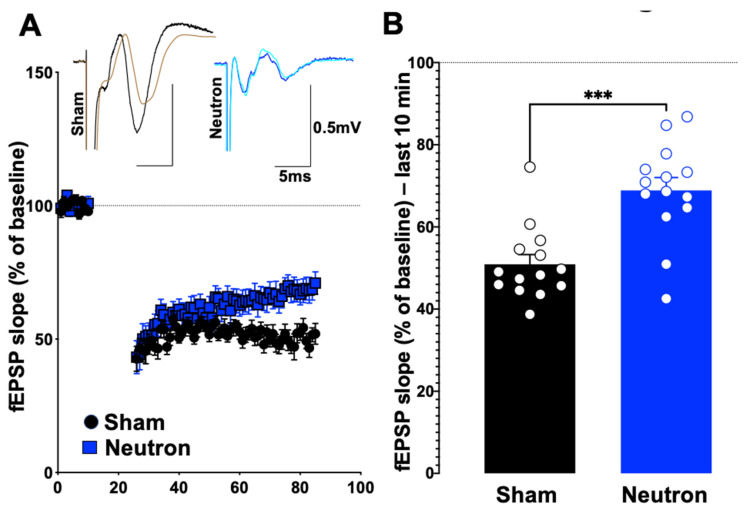
Chronic low dose neutron irradiation shows significant deficits in the low frequency stimulation associated long-term depression. (**A**) Stimulation of layer 2/3 and recording from the synapses to layer 5 using a low frequency stimulation (900 *×* 1 Hz) was performed in irradiated (blue circles) and age-matched sham mice (black circles). After a 10-min baseline, the LFS was performed for 15 min, following which LTD was measured for 60 min and plotted as fEPSP slope. (**B**) LFS-LTD (measured as average of last 10 min) in chronic low dose neutron irradiated mice (blue bar, 68.8 ± 3.137, *n* = 14) was significantly decreased (*** *p* = 0.0006, non-parametric, unpaired *t*-test, Mann–Whitney test) compared to control (black bar, 50.90 ± 2.375, *n* = 14). Representative traces are provided along with where the black trace show the pre-stimulation trace for control, while brown shows the post-stimulation trace. Blue shows the pre-, while pale green trace shows the post-stimulation trace for neutron-irradiated group.

**Figure 2 ijms-22-03668-f002:**
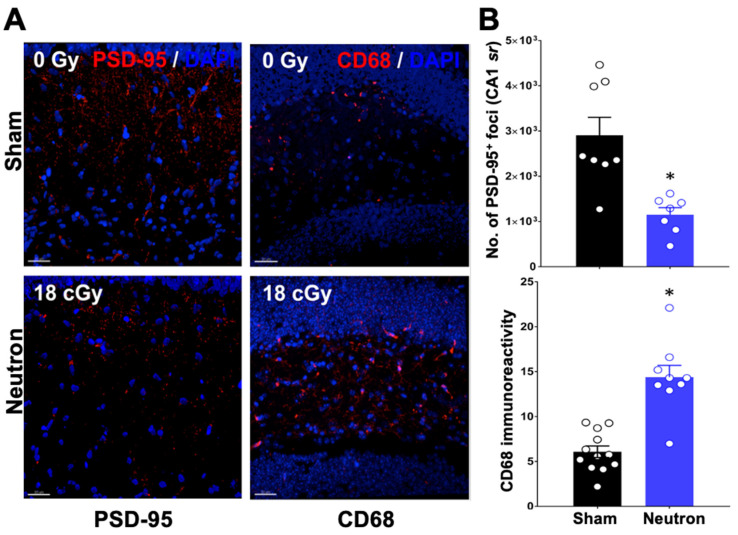
PSD-95 decreases while CD68 immunofluorescence staining increases in the dentate gyrus region of the mouse brain after three and a half months following neutron irradiation. (**A**) Representative staining of PSD-95 (red) in sham (0 Gy) counterstained with DAPI in blue is shown in the top left panel while the bottom right panel shows PSD-95 and DAPI in neutron irradiated brain region (18 Gy). CD-68 (red) in sham (0 Gy) is shown in the top right panel while the bottom right panel shows a representative staining in the neutron-irradiated brain region. (**B**) Volumetric quantification of PSD-95^+^ puncta (left panel) demonstrate significant reductions in molecular layer PSD-95 of the neutron-irradiated brain while quantitative analysis of CD68^+^ cells (right panel) demonstrates that compared to controls, irradiated mice exhibit increased microglial activation. Data presented as mean ± SEM where *n* = 2 tissues from each of 4 mice/group. *p* values derived from unpaired Student’s *t* tests. * *p* < 0.05. Scale bar = 30 mM.

**Figure 3 ijms-22-03668-f003:**
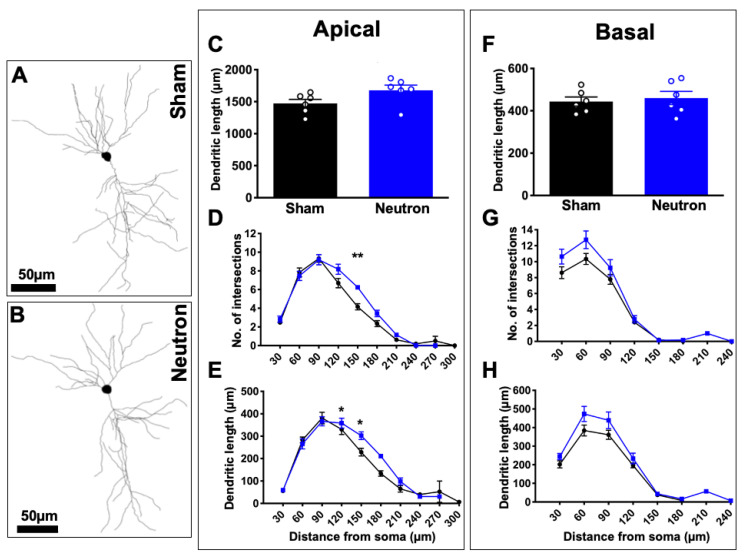
Apical neuronal complexity altered in response to chronic low-dose irradiation. Representative cell tracings from sham (**A**) and neutron (**B**) exposed mice. Scale bar = 50 mm. (**C**) Analysis of apical dendritic length. (**D**,**E**) Sholl analyses of apical dendrites. (**F**) Analysis of basal dendritic length. (**G**,**H**) Sholl analyses of basal dendrites. * *p* < 0.05; ** *p* < 0.01. Data represents mean ± SEM.

**Figure 4 ijms-22-03668-f004:**
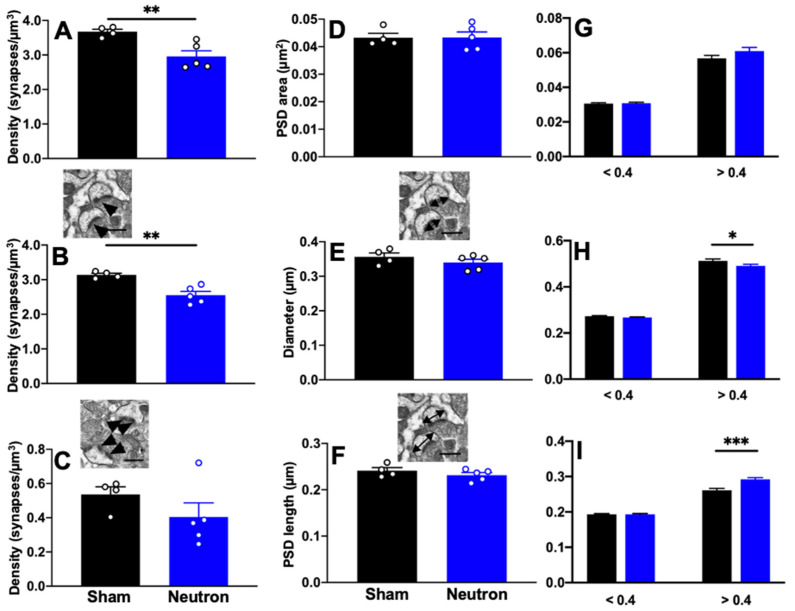
Synapse density is decreased in response to chronic low dose neutron-irradiation. There is a significant decrease in (**A**) total and (**B**) nonperforated synapse density in response to neutron irradiation. (**C**) No significant differences were observed in the density of perforated synapses. (**D**) Quantification of synapse morphology for all synapses revealed no differences in PSD area, PSD length or spine head diameter. However, when we looked at only nonperforated synapses and separated them according to head diameters into small (thin) and large (mushroom) spines, the lack of difference was still present in the PSD area (**G**) we found a significant decrease in larger non-perforated synapses in irradiated mice (**H**) but an increase in PSD length in irradiated mice compared to Sham (**I**). Inset images depict nonperforated and perforated synapses (arrowheads) in (**B**,**C**), respectively. Insets in (**E**,**F**) depict PSD and head diameter measurements. Unpaired *t*-test * *p* < 0.01, ** *p* < 0.001, *** *p* < 0.0001. *n* = 4 Sham and *n* = 5 Neutron mice/group. Data represents mean ± SEM.

**Figure 5 ijms-22-03668-f005:**
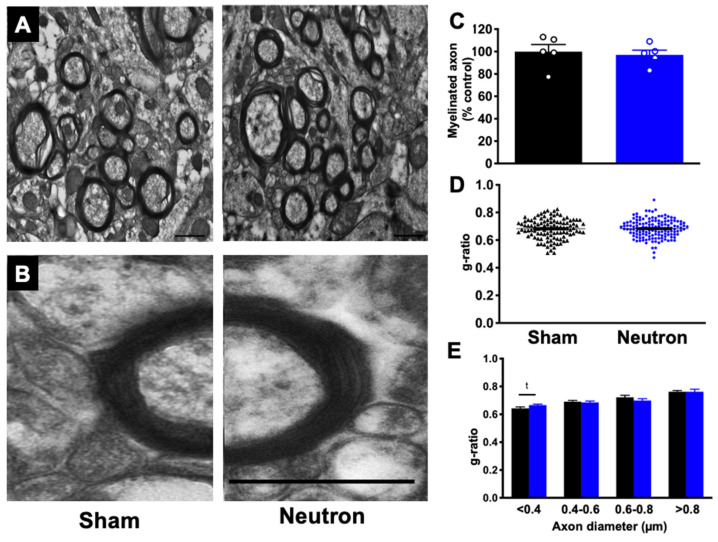
Chronic low dose neutron irradiation does not affect myelination. (**A**) Representative myelin images from sham and neutron treated mice. Scale bar = 800 nm. (**B**) Electron micrograph depicting a single axon of similar diameter in sham mice (**left**) and neutron irradiated mice (**right**). Scale bar = 400 nm. (**C**) There was no significant difference in the percentage of myelinated axons between neutron irradiated mice and sham. Data represents group means ± SEM. There is no difference in overall g-ratios in neutron irradiated mice compared to shams (**D**) and when binned according to axon diameter (**E**). There was an increase in g-ratio in axons with diameters < 0.4 um in irradiated mice, but it did not reach significance. Unpaired *t*-test; *p* = 0.058. Data represents individual measurements ± SEM.

**Figure 6 ijms-22-03668-f006:**
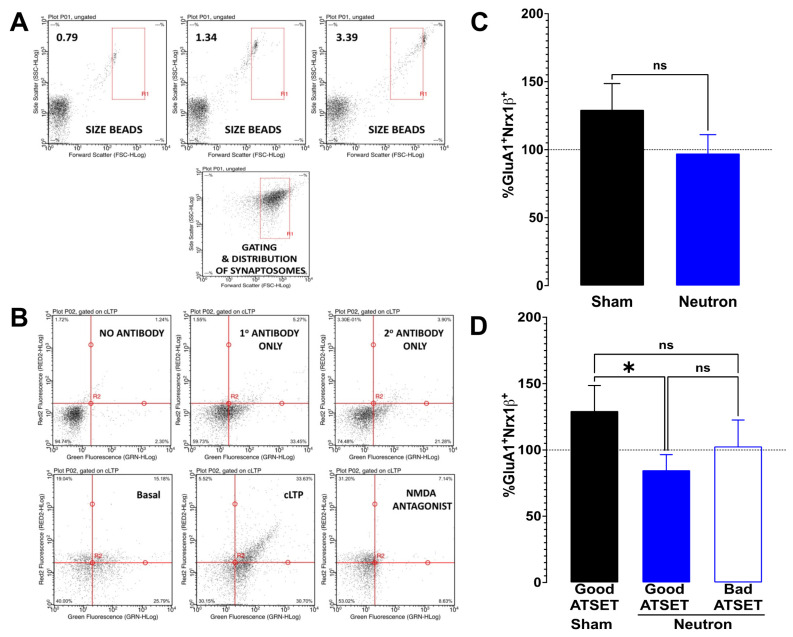
Fluorescence single synaptosome long-term potentiation (FASS-LTP) in chronic low-dose neutron irradiated rats. In order to estimate the functional status of the glutamate neurotransmission, hippocampal crude synaptosomal fractions were obtained from rats that were exposed chronic low dose neutron irradiation (black bars) and compared with age-matched sham controls (blue bars). (**A**) Representative forward scatter (FSC-HLog, x-axis) vs. side scatter (SSC-HLog, y-axis) dot plot showing the size–complexity profile of gated particles (inside R1 rectangle, size-gated synaptosomes). FSC–SSC dot plots using 0.79, 1.34 & 3.39 μm calibrated beads. (**B**) Representative plots showing the profile of scatter under different conditions. The numbers used in the estimation of cLTP are the upper right (UR) quadrant, which is showing (in%), the number of closely associated GluA1^+^Nrx1β^+^ particles gated to the synaptosomal size in that quadrant. Along the y-axis is the far-red fluorescence while along the x-axis is the green fluorescence that are couple to GluA1 and Nrx1β, respectively. The lower left quadrant represents the particles that do not show any appreciable association between the two proteins, while the lower right shows the particles scattered by Nrx1β and the upper left shows GluA1. Appreciable numbers of GluA1^+^Nrx1β^+^ scatter in the UR quadrant is only seen in the panel labeled “cLTP (chemically induced LTP)”. The panels that contain either no antibody or any one of the antibodies or extracellular buffer (basal) do not show a significant increase. The use of AP5, the NMDA antagonist, prevents the cLTP and is an effective negative control used in validating the response. (**C**) Potentiation levels in sham animals as a function of the basal response is shown in black bars (129.4 ± 19.18, *n* = 10) while the irradiated group is shown in blue bars (97.86 ± 13.87, *n* = 21); there is no significant difference between the two (*p* = 0.1917, unpaired *t*-test) (**D**) Upon separation of the irradiated groups into good performers vs. bad performers in the ATSET behavioral paradigm, the good performers (filled blue bars, 86.24 ± 13.45, *n* = 7) show a significant difference (* *p* = 0.044) compared to the sham, but the bad performers (clear blue bars, 102.7 ± 19.88, *n* = 14) do not show any significant difference to either groups.

**Figure 7 ijms-22-03668-f007:**
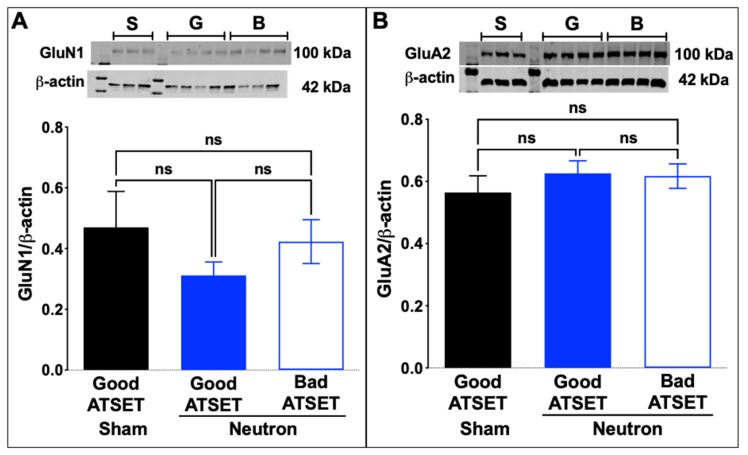
Western blot analyses for the ionotropic glutamate receptors in the sham and the low dose neutron irradiated groups of rats. (**A**) No significant differences were appreciated in the GluN1 subunit of the NMDA receptor between the three groups of rats (**B**) No significant change was observed in the expression levels of the GluA2 subunit of the AMPAR between the three groups of rats. Crude synaptosomal fractions from three-four animals per group were used in the assessment and statistical significance (*p* < 0.05) was determined using non-parametric, Kruskal–Wallis *t*-test.

**Figure 8 ijms-22-03668-f008:**
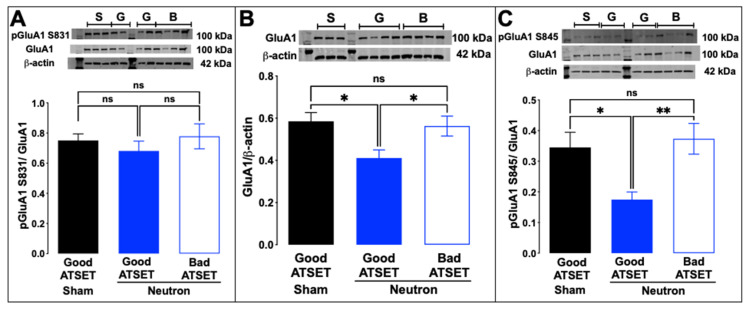
Phosphorylation of serine at position 845 of the GluA1 subunit is significantly decreased in good ATSET rats. (**A**) The phosphorylation state of the serine residue at position 831 was not altered, but (**C**) at position 845, the good ATSET group showed lower phosphorylation of serine compared to the other two groups. (**B**) Interestingly, significantly lower expression of GluA1 subunit of the AMPA receptors was noted in the good ATSET neutron rats compared to the other two groups. Crude synaptosomal fractions from three-four animals per group were used in the assessment and statistical significance (* *p* < 0.05, ** *p* < 0.01) was determined using non-parametric, Kruskal–Wallis *t*-test.

**Figure 9 ijms-22-03668-f009:**
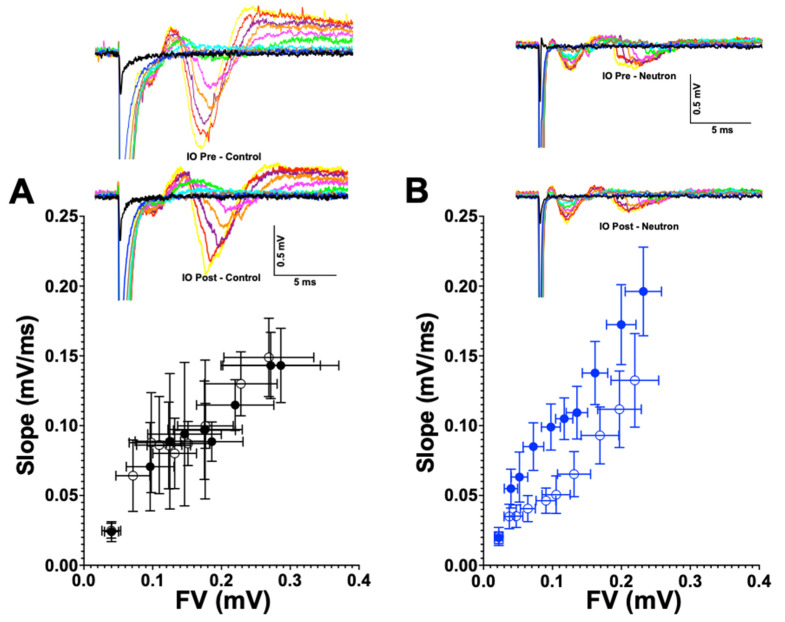
Input-output curves are not altered between the control and the chronic low dose neutron-irradiated group. (**A**) Pre-(filled circles) and post-LFS input-output relationships (clear circles) for the control group are plotted as a function of slope (mV/ms) on the y-axis and the fiber volley (mV) along the X-axis to estimate whether there are changes in the synaptic strength of the response, which would be observed in the either the leftward or rightward shift of the post curves. Representative curves are provided adjacently, where increasing stimulation currents are plotted in different colors. (**B**) Pre (filled blue circles) and post-LFS (clear blue circles) input-output relationship for the neutron irradiated group do not show significant differences. Representative curves are presented alongside, similar to the control group.

**Figure 10 ijms-22-03668-f010:**
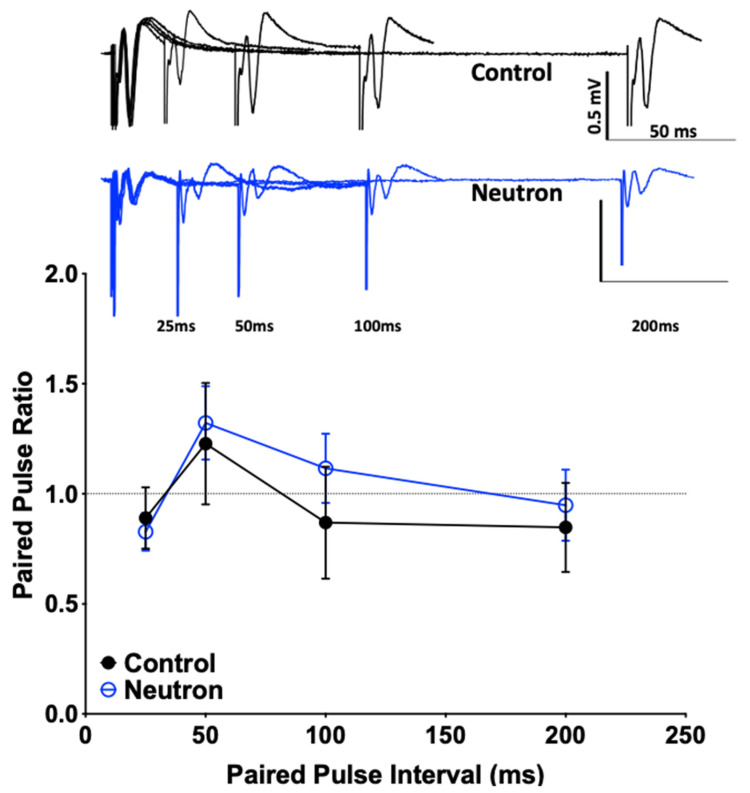
Paired pulse ratio is not significantly different between the control and the chronic low dose neutron-irradiated group. Representative traces for the control (black) and the neutron irradiated group (blue) are provided. There is no significant difference in the paired pulse ratio (calculated as the fEPSP slope of second pulse over the first) at four different intervals tested (25, 50, 100 and 200 ms), suggesting that there is no presynaptic deficit that accounts for the effects of the irradiation associates synaptic or behavioral changes.

**Table 1 ijms-22-03668-t001:** Values represent the mean ± standard error of mean for *n* = 14 animals, calculated as the ratio of fEPSP slope values of the second response over the first response averaged over thirty sweeps. The assessment was conducted both before low frequency stimulation before baseline (LFS = 900 stimulations of 1 Hz, pre-LFS) and 60 min following LFS (post-LFS). No significant differences were observed at any of the above intervals.

Paired Pulse Interval (ms)	Pre-LFS		Post-LFS	
	Sham	Neutron	Sham	Neutron
200	0.847 ± 0.202	0.948 ± 0.162	0.479 ± 0.130	0.497 ± 0.267
100	0.869 ± 0.254	1.116 ± 0.158	0.732 ± 0.202	0.607 ± 0.111
50	1.227 ± 0.276	1.322 ± 0.167	0.980 ± 0.295	1.112 ± 0.145
25	0.890 ± 0.139	0.827 ± 0.086	0.510 ± 0.224	0.672 ± 0.159

## Data Availability

Data will be made available in the NASA publication repository PubSpace (https://www.ncbi.nlm.nih.gov/pmc/funder/nasa/). Accessed on 31 March 2021.

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
