# Peer review of "Chronic Low Dose Neutron Exposure Results in Altered Neurotransmission Properties of the Hippocampus-Prefrontal Cortex Axis in Both Mice and Rats"

_ijms, 2021, doi:10.3390/ijms22073668_

Round 1
Reviewer 1 Report
The authors investigated how chronic low dose rate neutron irradiation influence the homeostatic synaptic plasticity in mice and rats. By using multiple experimental paradigms, the authors analyzed the electrophysiological, structural, molecular changes in neutron-irradiated animals. The experiments are well-designed and the results are solid.
Major concerns:
- Could the authors specify the age of the mice when they were started to receive neutron irradiation? Will the mice become to old for LTP or LTD analysis after chronic neutron irradiation? Does the ages of animals mimic the human subjects?
- Only male animals were used in this study, could gender produce different results?
Author Response
The authors investigated how chronic low dose rate neutron irradiation influence the homeostatic synaptic plasticity in mice and rats. By using multiple experimental paradigms, the authors analyzed the electrophysiological, structural, molecular changes in neutron-irradiated animals. The experiments are well-designed and the results are solid.
Major concerns:
- Could the authors specify the age of the mice when they were started to receive neutron irradiation?
On line 626, we have specified that the age of the mice was six months when they started receiving neutron irradiation.
- Will the mice become too old for LTP or LTD analysis after chronic neutron irradiation? Does the ages of animals mimic the human subjects?
According to Flurkey, Currer and Harrison (2007 – https://doi.org/10.1016/B978-012369454-6/50074-1), the age of the mice at 16-18 months is equivalent with mid 40s to mid 50s human age.
- Only male animals were used in this study, could gender produce different results?
While an important psychosocial term for humans where gender denotes the individual’s (or community’s) perception of an individual’s sexuality, there is no understanding of whether rodents perceive their sexuality in a similar manner. However, it is important to address the sexual dependency of space radiation. Our observation is that studies to date, which are, in no measure, comprehensive, provide some highly context-specific behavior (a couple of mouse strains), but without enough evidence one way or another (low numerator effects caused inferior sham performance in females). Thus, at the present stage, it remains unknown if female rodents would show more or less effect and would be an important factor to be explored in future studies.
Reviewer 2 Report
Krishnan et al presented the effects for neural properties after chronic low dose neutron irradiation to rodents. Irradiation condition simulated space mission of NASA and astronauts radiation exposure. The manuscript is very interesting and authors presented many data. I have no problem for this manuscript except figure qualities. Generally, figure fonts in figure and legends are too small. I had very hard time to see. I appreciate authors presented each data point along with bar graphs. But some figures are difficult to see due to their size or color. It is same thing for font size. I am afraid this manuscript does not follow journal's format. Please follow journal's instruction.
Author Response
We appreciate the constructive criticism, we have now included figures separately from captions, and making changes in the figures to increase visibility. Additionally, we have provided a zip file with all the individual figures in TIFF format as specified in the author’s instructions.